# DIFFERENTIAL TRANSFORMER

**Tianzhu Ye**[* ‡†]    **Li Dong**[* †]    **Yuqing Xia**[* †]    **Yutao Sun**[* ‡†]
**Yi Zhu**[†]    **Gao Huang**[‡◇]    **Furu Wei**[†◇]
[‡] BNRist, Tsinghua University, Beijing, China      [†] Microsoft Research
https://aka.ms/GeneralAI

## ABSTRACT

Transformer tends to overallocate attention to irrelevant context. In this work, we introduce DIFF Transformer, which amplifies attention to the relevant context while canceling noise. Specifically, the differential attention mechanism calculates attention scores as the difference between two separate softmax attention maps. The subtraction cancels noise, promoting the emergence of sparse attention patterns. Experimental results on language modeling show that DIFF Transformer outperforms Transformer in various settings of scaling up model size and training tokens. More intriguingly, it offers notable advantages in practical applications, such as long-context modeling, key information retrieval, hallucination mitigation, in-context learning, and reduction of activation outliers. By being less distracted by irrelevant context, DIFF Transformer can mitigate hallucination in question answering and text summarization. For in-context learning, DIFF Transformer not only enhances accuracy but is also more robust to order permutation, which was considered as a chronic robustness issue. The results position DIFF Transformer as a highly effective and promising architecture to advance large language models.

## 1 INTRODUCTION

Transformer (Vaswani et al., 2017) has garnered significant research interest in recent years, with the decoder-only Transformer emerging as the de facto standard for large language models (LLMs). At the heart of Transformer is the attention mechanism, which employs the softmax function to weigh the importance of various tokens in a sequence. However, recent studies (Kamradt, 2023; Liu et al., 2024b) show that LLMs face challenges in accurately retrieving key information from context.

As illustrated on the left side of Figure 1, we visualize the normalized attention scores assigned to different parts of the context by a Transformer. The task is to retrieve an answer embedded in the middle of a pile of documents. The visualization reveals that Transformer tends to allocate only a small proportion of attention scores to the correct answer, while disproportionately focusing on irrelevant context. The experiments in Section 3 further substantiate that Transformers struggle with such capabilities. The issue arises from non-negligible attention scores assigned to irrelevant context, which ultimately drowns out the correct answer. We term these extraneous scores as *attention noise*.

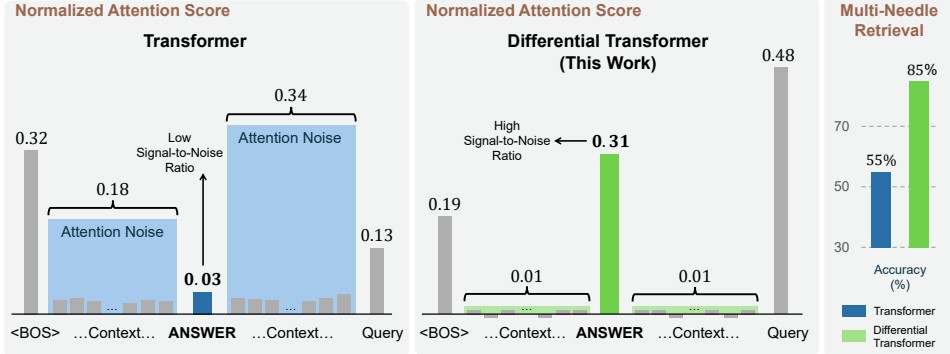

Figure 1: Transformer often over-attends to irrelevant context (i.e., attention noise). DIFF Transformer amplifies attention to answer spans and cancels noise, enhancing the capability of context modeling.

---

* Equal contribution. ◇ Corresponding authors.

In this paper, we introduce Differential Transformer (a.k.a. DIFF Transformer), a foundation architecture for large language models. The differential attention mechanism is proposed to cancel attention noise with differential denoising. Specifically, we partition the query and key vectors into two groups and compute two separate softmax attention maps. Then the result of subtracting these two maps is regarded as attention scores. The differential attention mechanism eliminates attention noise, encouraging models to focus on critical information. The approach is analogous to noise-canceling headphones and differential amplifiers (Laplante et al., 2018) in electrical engineering, where the difference between two signals cancels out common-mode noise. In the middle of Figure 1, we also present the normalized distribution of attention scores for DIFF Transformer. We observe that DIFF Transformer assigns significantly higher scores to the correct answer and much lower scores to irrelevant context compared to Transformer. The right side of Figure 1 shows that the proposed method achieves notable improvements in retrieval capability.

We conduct extensive experiments on language modeling. We scale up DIFF Transformer in terms of parameter count, training tokens, and context length. The scaling curves indicate that DIFF Transformer requires only about 65% of model size or training tokens needed by Transformer to achieve comparable language modeling performance. Moreover, DIFF Transformer outperforms Transformer in various downstream tasks. The long-sequence evaluation also shows that DIFF Transformer is highly effective in utilizing the increasing context. In addition, the experimental results demonstrate that DIFF Transformer has intriguing advantages for large language models. For example, the proposed method substantially outperforms Transformer in key information retrieval, hallucination mitigation, in-context learning, and mathematical reasoning. DIFF Transformer also reduces outliers in model activations, which provides new opportunities for quantization. The findings establish DIFF Transformer as an effective and distinctive foundation architecture for large language models.

## 2 DIFFERENTIAL TRANSFORMER

We propose Differential Transformer (a.k.a. DIFF Transformer) as a foundation architecture for sequence modeling, such as large language models (LLMs). We take a decoder-only model as an example to describe the architecture. The model is stacked with $L$ DIFF Transformer layers. Given an input sequence $x = x_1 \cdots x_N$, we pack the input embeddings into $X^0 = [\boldsymbol{x}_1, \cdots, \boldsymbol{x}_N] \in \mathbb{R}^{N \times d_{\text{model}}}$, where $d_{\text{model}}$ represents the hidden dimension of the model. The input is further contextualized to obtain the output $X^L$, i.e., $X^l = \text{Decoder}(X^{l-1})$, $l \in [1, L]$. Each layer consists of two modules: a differential attention module followed by a feed-forward network module. Compared to Transformer (Vaswani et al., 2017), the main difference is the replacement of conventional softmax attention with differential attention while the macro layout is kept the same. We also adopt pre-RMSNorm (Zhang & Sennrich, 2019) and SwiGLU (Shazeer, 2020; Ramachandran et al., 2017) as improvements following LLaMA (Touvron et al., 2023).

### 2.1 DIFFERENTIAL ATTENTION

The differential attention mechanism maps query, key, and value vectors to outputs. We use query and key vectors to compute attention scores, and then compute a weighted sum of value vectors. The critical design is that we use a pair of softmax functions to cancel the noise of attention scores. Specifically, given input $X \in \mathbb{R}^{N \times d_{\text{model}}}$, we first project them to query, key, and value $Q_1, Q_2, K_1, K_2 \in \mathbb{R}^{N \times d}$, $V \in \mathbb{R}^{N \times 2d}$. Then the differential attention operator $\text{DiffAttn}(\cdot)$ computes outputs via:

$$[Q_1; Q_2] = XW^Q, \quad [K_1; K_2] = XW^K, \quad V = XW^V$$
$$\text{DiffAttn}(X) = (\text{softmax}(\frac{Q_1 K_1^T}{\sqrt{d}}) - \lambda \, \text{softmax}(\frac{Q_2 K_2^T}{\sqrt{d}}))V \tag{1}$$

where $W^Q, W^K, W^V \in \mathbb{R}^{d_{\text{model}} \times 2d}$ are parameters, and $\lambda$ is a learnable scalar. In order to synchronize the learning dynamics, we re-parameterize the scalar $\lambda$ as:

$$\lambda = \exp(\lambda_{\mathbf{q_1}} \cdot \lambda_{\mathbf{k_1}}) - \exp(\lambda_{\mathbf{q_2}} \cdot \lambda_{\mathbf{k_2}}) + \lambda_{\text{init}} \tag{2}$$

where $\lambda_{\mathbf{q_1}}, \lambda_{\mathbf{k_1}}, \lambda_{\mathbf{q_2}}, \lambda_{\mathbf{k_2}} \in \mathbb{R}^d$ are learnable vectors, and $\lambda_{\text{init}} \in (0, 1)$ is a constant used for the initialization of $\lambda$. We empirically find that the setting $\lambda_{\text{init}} = 0.8 - 0.6 \times \exp(-0.3 \cdot (l-1))$ works

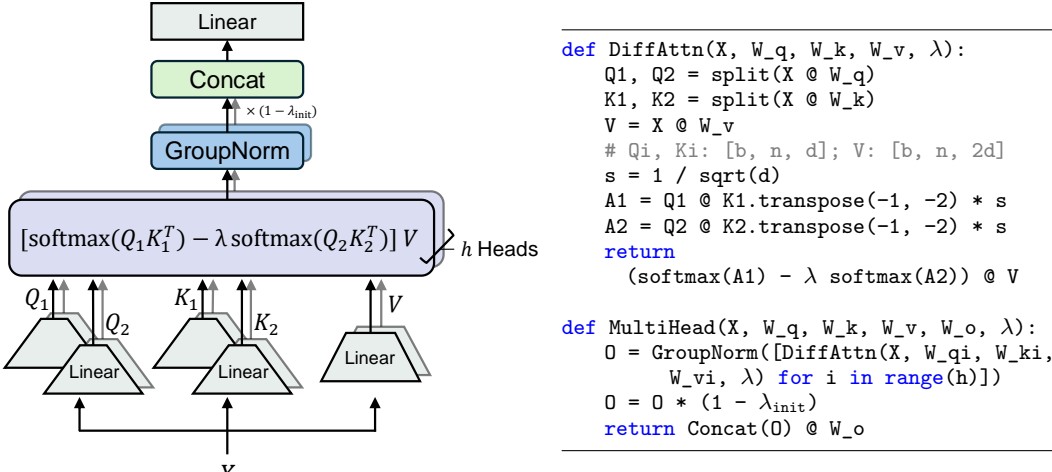

```
def DiffAttn(X, W_q, W_k, W_v, λ):
    Q1, Q2 = split(X @ W_q)
    K1, K2 = split(X @ W_k)
    V = X @ W_v
    # Qi, Ki: [b, n, d]; V: [b, n, 2d]
    s = 1 / sqrt(d)
    A1 = Q1 @ K1.transpose(-1, -2) * s
    A2 = Q2 @ K2.transpose(-1, -2) * s
    return
        (softmax(A1) - λ softmax(A2)) @ V

def MultiHead(X, W_q, W_k, W_v, W_o, λ):
    O = GroupNorm([DiffAttn(X, W_qi, W_ki,
        W_vi, λ) for i in range(h)])
    O = O * (1 - λ_init)
    return Concat(O) @ W_o
```

Figure 2: Multi-head differential attention. Each head takes the difference between two $\mathrm{softmax}$ attention maps to cancel out attention noise. $\lambda$ is a learnable scalar that is initialized to $\lambda_{\mathrm{init}}$. GroupNorm applies normalization to each head independently. A fixed multiplier $(1 - \lambda_{\mathrm{init}})$ is used after GroupNorm, which aligns the gradient flow with Transformer. The code implementation is available at https://aka.ms/Diff-Transformer.

well in practice, where $l \in [1, L]$ represents layer index. It is used as the default strategy in our experiments. We also explore using the same $\lambda_{\mathrm{init}}$ (e.g., 0.8) for all layers as another initialization strategy. As shown in the ablation studies (Section 3.8), the performance is relatively robust to different initialization strategies.

Differential attention takes the difference between two $\mathrm{softmax}$ attention functions to eliminate attention noise. The idea is analogous to differential amplifiers (Laplante et al., 2018) proposed in electrical engineering, where the difference between two signals is used as output, so that we can null out the common-mode noise of the input. Naderi et al. (2024) also prove that differential attention makes the spectral distribution of attention matrices more balanced, which effectively resolves rank collapse. In addition, the design of noise-canceling headphones is based on a similar idea. We can directly reuse FlashAttention (Dao et al., 2022) as described in Appendix A, which significantly improves model efficiency.

**Multi-Head Differential Attention** We also use the multi-head mechanism (Vaswani et al., 2017) in Differential Transformer. Let $h$ denote the number of attention heads. We use different projection matrices $W_i^Q, W_i^K, W_i^V, i \in [1, h]$ for the heads. The scalar $\lambda$ is shared between heads within the same layer. Then the head outputs are normalized and projected to the final results as follows:

$$\mathrm{head}_i = \mathrm{DiffAttn}(X; W_i^Q, W_i^K, W_i^V, \lambda)$$
$$\overline{\mathrm{head}_i} = (1 - \lambda_{\mathrm{init}}) \cdot \mathrm{LN}(\mathrm{head}_i) \tag{3}$$
$$\mathrm{MultiHead}(X) = \mathrm{Concat}(\overline{\mathrm{head}_1}, \cdots, \overline{\mathrm{head}_h}) W^O$$

where $\lambda_{\mathrm{init}}$ is the constant scalar in Equation (2), $W^O \in \mathbb{R}^{d_{\mathrm{model}} \times d_{\mathrm{model}}}$ is a learnable projection matrix, $\mathrm{LN}(\cdot)$ uses RMSNorm (Zhang & Sennrich, 2019) for each head, and $\mathrm{Concat}(\cdot)$ concatenates the heads together along the channel dimension. We use a fixed multiplier $(1 - \lambda_{\mathrm{init}})$ as the scale of $\mathrm{LN}(\cdot)$ to align the gradients with Transformer. Appendix G proves that the overall gradient flow remains similar to that of Transformer. The nice property enables us to directly inherit similar hyperparameters and ensures training stability. We set the number of heads $h = d_{\mathrm{model}}/2d$, where $d$ is equal to the head dimension of Transformer. So we can align the parameter counts and computational complexity.

**Headwise Normalization** Figure 2 uses $\mathrm{GroupNorm}(\cdot)$ (Wu & He, 2018) to emphasize that $\mathrm{LN}(\cdot)$ is applied to each head independently. As differential attention tends to have a sparser pattern, statistical information is more diverse between heads. The $\mathrm{LN}(\cdot)$ operator normalizes each head before concatenation to improve gradient statistics (Wang et al., 2023; Qin et al., 2022).

## 2.2 OVERALL ARCHITECTURE

The overall architecture stacks $L$ layers, where each layer contains a multi-head differential attention module, and a feed-forward network module. We describe the Differential Transformer layer as:

$$Y^l = \text{MultiHead}(\text{LN}(X^l)) + X^l \tag{4}$$

$$X^{l+1} = \text{SwiGLU}(\text{LN}(Y^l)) + Y^l \tag{5}$$

where $\text{LN}(\cdot)$ is RMSNorm (Zhang & Sennrich, 2019), $\text{SwiGLU}(X) = (\text{swish}(XW^G) \odot XW_1)W_2$, and $W^G, W_1 \in \mathbb{R}^{d_{model} \times \frac{8}{3} d_{model}}, W_2 \in \mathbb{R}^{\frac{8}{3} d_{model} \times d_{model}}$ are learnable matrices.

## 3 EXPERIMENTS

We evaluate Differential Transformer for large language models from the following perspectives. First, we compare the proposed architecture with Transformers in various downstream tasks (Section 3.1) and study the properties of scaling up model size and training tokens (Section 3.2). Second, we conduct a length extension to 64K and evaluate the long-sequence modeling capability (Section 3.3). Third, we present the results of key information retrieval, contextual hallucination evaluation, and in-context learning (Sections 3.4–3.6). Forth, we show that Differential Transformer can reduce outliers in the model activations compared to Transformer (Section 3.7). Fifth, we conduct extensive ablation studies for various design choices (Section 3.8).

## 3.1 LANGUAGE MODELING EVALUATION

We train 3B-size DIFF Transformer language models on 1T tokens and compare with previous well-trained Transformer-based models (Geng & Liu, 2023; Tow, 2023; Tow et al., 2023) in various downstream tasks. As described in Appendix B, we follow the same setting to train a 3B-size Transformer language model on 350B tokens. The checkpoints are also used in the following experiments and analysis to ensure fair comparisons.

**Setup** We follow a similar recipe as StableLM-3B-4E1T (Tow et al., 2023). We set hidden size to 3072. The number of layers is 28. The head dimension $d$ is 128. The number of heads is 24 for Transformer and 12 for DIFF Transformer, to align computation FLOPs and model size. The total parameter count is about 2.8B. The training sequence length is 4096. The batch size is 4M tokens. We train the models with 1T tokens. We use AdamW (Loshchilov & Hutter, 2019) optimizer with $\beta = 0.9, 0.95$. The maximal learning rate is 3.2e-4 with 1000 warmup steps and linearly decays to 1.28e-5. The training corpus also follows StableLM-3B-4E1T (Tow et al., 2023). We employ `tiktoken-cl100k_base` tokenizer. Detailed hyperparameters are provided in Appendix D.

**Results** Table 1 reports the zero-shot results on the LM Eval Harness benchmark (Gao et al., 2023). We compare DIFF Transformer with well-trained Transformer-based language models, including OpenLLaMA-v2-3B (Geng & Liu, 2023), StableLM-base-alpha-3B-v2 (Tow, 2023), and StableLM-3B-4E1T (Tow et al., 2023). OpenLLaMA-v2-3B and StableLM-base-alpha-3B-v2 are also trained with 1T tokens. The 1T results of StableLM-3B-4E1T are taken from its technical report (Tow et al., 2023). Experimental results show that DIFF Transformer achieves favorable performance compared to previous well-tuned Transformer language models. In addition, Appendix B shows that DIFF Transformer outperforms Transformer across various tasks, where we use the same setting to train the 3B-size language models for fair comparisons.

| Model | ARC-C | ARC-E | BoolQ | HellaSwag | OBQA | PIQA | WinoGrande | Avg |
|---|---|---|---|---|---|---|---|---|
| *Training with 1T tokens* | | | | | | | | |
| OpenLLaMA-3B-v2 (Geng & Liu, 2023) | 33.9 | 67.6 | 65.7 | 70.0 | 26.0 | 76.7 | 62.9 | 57.5 |
| StableLM-base-alpha-3B-v2 (Tow, 2023) | 32.4 | 67.3 | 64.6 | 68.6 | 26.4 | 76.0 | 62.1 | 56.8 |
| StableLM-3B-4E1T (Tow et al., 2023) | — | 66.6 | — | — | — | **76.8** | 63.2 | — |
| DIFF-3B | **37.8** | **72.9** | **69.0** | **71.4** | **29.0** | 76.8 | **67.1** | **60.6** |

Table 1: Eval Harness Gao et al. (2023) accuracy compared with well-trained Transformer language models (Tow et al., 2023; Tow, 2023; Geng & Liu, 2023). We scale the 3B model to 1 trillion training tokens. The 1T results of StableLM-3B-4E1T are taken from its technical report Tow et al. (2023).

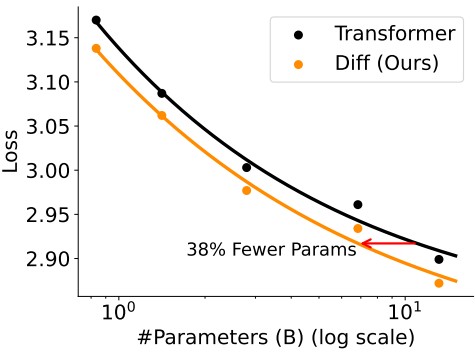
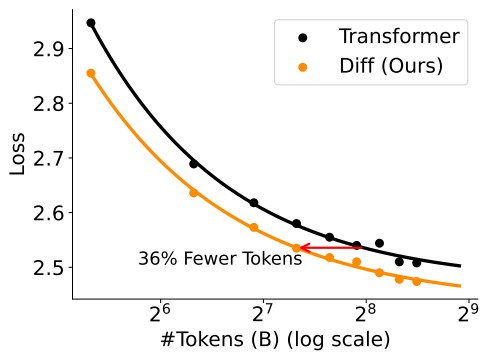

(a) Scaling model size ranging from 830M to 13B.  (b) Scaling number of training tokens for 3B models.

Figure 3: Language modeling loss of scaling up parameter count and training tokens. DIFF Transformer requires only about 65% of model size or training tokens to match Transformer's performance.

## 3.2 SCALABILITY COMPARED WITH TRANSFORMER

We compare the scaling properties of DIFF Transformer and Transformer on language modeling. We scale up the model size, and the number of training tokens, respectively. We follow the augmented Transformer architecture as in LLaMA (Touvron et al., 2023) and use the same setting to ensure fair comparison. Specifically, the "Transformer" models include improvements in RMSNorm (Zhang & Sennrich, 2019), SwiGLU (Shazeer, 2020; Ramachandran et al., 2017), and removal of bias.

**Scaling Model Size**  As shown in Figure 3a, we train language models with 830M, 1.4B, 2.8B, 6.8B, and 13.1B parameters. The models are trained with a sequence length of 2048, and a batch size of 0.25M tokens. We train models for 40K steps. Detailed hyperparameters are described in Appendix E. The scaling law (Kaplan et al., 2020) empirically fits well in this configuration. Figure 3a shows that DIFF Transformer outperforms Transformer in various model sizes. The results indicate that DIFF Transformer is scalable in terms of parameter count. According to the fitted curves, 6.8B-size DIFF Transformer achieves a validation loss comparable to 11B-size Transformer, requiring only **62.2%** of parameters. Similarly, 7.8B-size DIFF Transformer matches the performance of 13.1B-size Transformer, requiring only **59.5%** of parameters.

**Scaling Training Tokens**  As shown in Figure 3b, we evaluate the 3B language models (as presented in Appendix B) every 40B tokens (i.e., 10K steps) up to a total of 360B tokens (i.e., 90K steps). The fitted curves indicate that DIFF Transformer trained with 160B tokens achieves comparable performance as Transformer trained with 251B tokens, consuming only **63.7%** of the training tokens.

## 3.3 LONG-CONTEXT EVALUATION

We extend the 3B-size language models (described in Appendix B) to 64K context length. We continue training the 3B checkpoints for additional 1.5B tokens. Most hyperparameters are kept the same as in Section 3.1. The learning rate is 8e-5. The RoPE (Su et al., 2021) $\theta$ is increased to 640,000. The training corpus is up-sampled according to sequence length (Fu et al., 2024).

**Results**  Figure 4 presents cumulative average negative log-likelihood (NLL) of the tokens at varying positions (Reid et al., 2024), where lower NLL indicates better performance. The evaluation is conducted on book data within 64K length. We observe a consistent decrease in NLL as the context length increases. DIFF Transformer achieves lower NLL values than Transformer. The results demonstrate that DIFF Transformer can effectively leverage the increasing context.

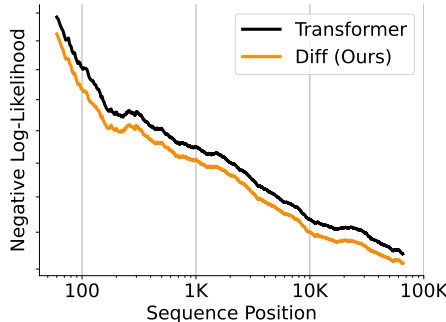

Figure 4: Cumulative average negative log-likelihood (lower is better) on book data. DIFF Transformer leverages long context more effectively.

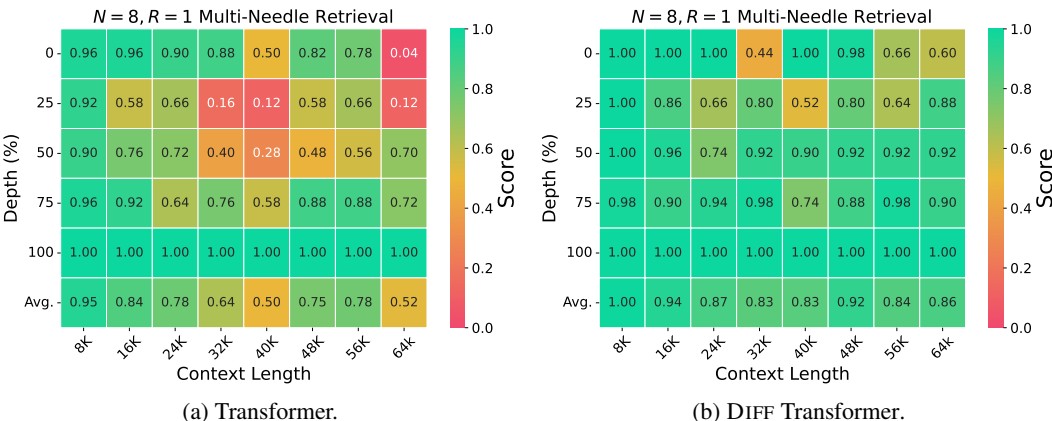

Figure 5: Multi-needle retrieval results in 64k length.

## 3.4 KEY INFORMATION RETRIEVAL

The Needle-In-A-Haystack (Kamradt, 2023) test is widely used to evaluate the ability to extract critical information embedded in a large context. We follow the multi-needle evaluation protocol of LWM (Liu et al., 2024a) and Gemini 1.5 (Reid et al., 2024). The needles are inserted into varying depths within contexts of different lengths. Each needle consists of a concise sentence that assigns a unique magic number to a specific city. The goal is to retrieve the magic numbers corresponding to the query cities. We position the answer needle at five different depths within the context: 0%, 25%, 50%, 75%, and 100%, while placing other distracting needles randomly. Each combination of depth and length is evaluated using 50 samples. The average accuracy is reported. Let $N$ denote the total number of number-city pairs and $R$ the number of query cities.

**Retrieve from 4K Context Length**   As shown in Table 2, we insert $N = 1, 2, 4, 6$ needles into 4K-length contexts and retrieve $R = 1, 2$ needles. We evaluate 3B-size models trained with 4K input length (Appendix B). We find that both models obtain good accuracy for $N = 1$ and $N = 2$. As $N$ and $R$ increase, DIFF Transformer maintains a consistent accuracy, while the performance of Transformer drops significantly. In particular, at $N = 6, R = 2$, the accuracy gap between the two models reaches 30%. The results indicate the superior ability of DIFF Transformer to retrieve key information in distracting contexts.

| Model | $N=1$ $R=1$ | $N=2$ $R=2$ | $N=4$ $R=2$ | $N=6$ $R=2$ |
|---|---|---|---|---|
| Transformer | **1.00** | 0.85 | 0.62 | 0.55 |
| DIFF | **1.00** | **0.92** | **0.84** | **0.85** |

Table 2: Multi-needle retrieval accuracy in 4K length, averaged over the answer needle positions. $N$ represents the number of needles, and $R$ denotes the number of query cities.

**Retrieve from 64K Context Length**   As shown in Figure 5, the evaluated context length ranges from 8K to 64K for the $N = 8, R = 1$ setting. We evaluate the 3B-size models with length extension (Section 3.3). We report the accuracy across varying answer needle depths (y-axis) and context lengths (x-axis). The bottom row is the average accuracy for all depths. DIFF Transformer maintains stable performance across different context lengths. In contrast, Transformer's average accuracy gradually declines as the context length increases up to the maximal length, i.e., 64K. Besides, DIFF Transformer outperforms Transformer particularly when key information is positioned within the first half of the context (i.e., 0%, 25%, and 50% depth). In particular, when needles are placed at the 25% depth in a 64K context, DIFF Transformer achieves 76% accuracy improvement over Transformer.

**Attention Score Analysis**   Table 3 presents the attention scores allocated to the answer span and the noise context for the key information retrieval task. The scores indicate the model's ability to preserve useful information against attention noise. We compare the normalized attention scores when key information is inserted at different positions (i.e., depths) within the context. Compared with Transformer, DIFF Transformer allocates higher attention scores to the answer span and has lower attention noise.

| Model | Attention to Answer ↑ | | | | | Attention Noise ↓ | | | | |
|---|---|---|---|---|---|---|---|---|---|---|
| | 0% | 25% | 50% | 75% | 100% | 0% | 25% | 50% | 75% | 100% |
| Transformer | 0.03 | 0.03 | 0.03 | 0.07 | 0.09 | 0.51 | 0.54 | 0.52 | 0.49 | 0.49 |
| DIFF | **0.27** | **0.30** | **0.31** | **0.32** | **0.40** | **0.01** | **0.02** | **0.02** | **0.02** | **0.01** |

Table 3: Attention scores allocated to answer spans and noise context in the key information retrieval task. The target answer is inserted in varying positions (i.e., depth) of context. DIFF Transformer allocates more attention scores to useful information and effectively cancels out attention noise.

## 3.5 IN-CONTEXT LEARNING

We evaluate in-context learning from two perspectives, including many-shot classification and robustness of in-context learning. In-context learning is a fundamental capability of language models, which indicates how well a model can utilize input context.

**Many-Shot In-Context Learning** As presented in Figure 6, we compare the accuracy of many-shot classification between Transformer and our architecture. We evaluate the 3B-size language models that support 64K input length (Section 3.3). We follow the evaluation protocol of (Bertsch et al., 2024) and use constrained decoding (Ratner et al., 2023). We incrementally increase the number of demonstration samples from 1-shot until the total length reaches 64K length. Specifically, the TREC (Hovy et al., 2001) dataset has 6 classes, TREC-fine (Hovy et al., 2001) has 50 classes, Banking-77 (Casanueva et al., 2020) has 77 classes, and Clinic-150 (Larson et al., 2019) has 150 classes. The results show that DIFF Transformer consistently outperforms Transformer across datasets and varying numbers of demonstration samples. Moreover, the improvement in average accuracy is substantial, ranging from 5.2% to 21.6%.

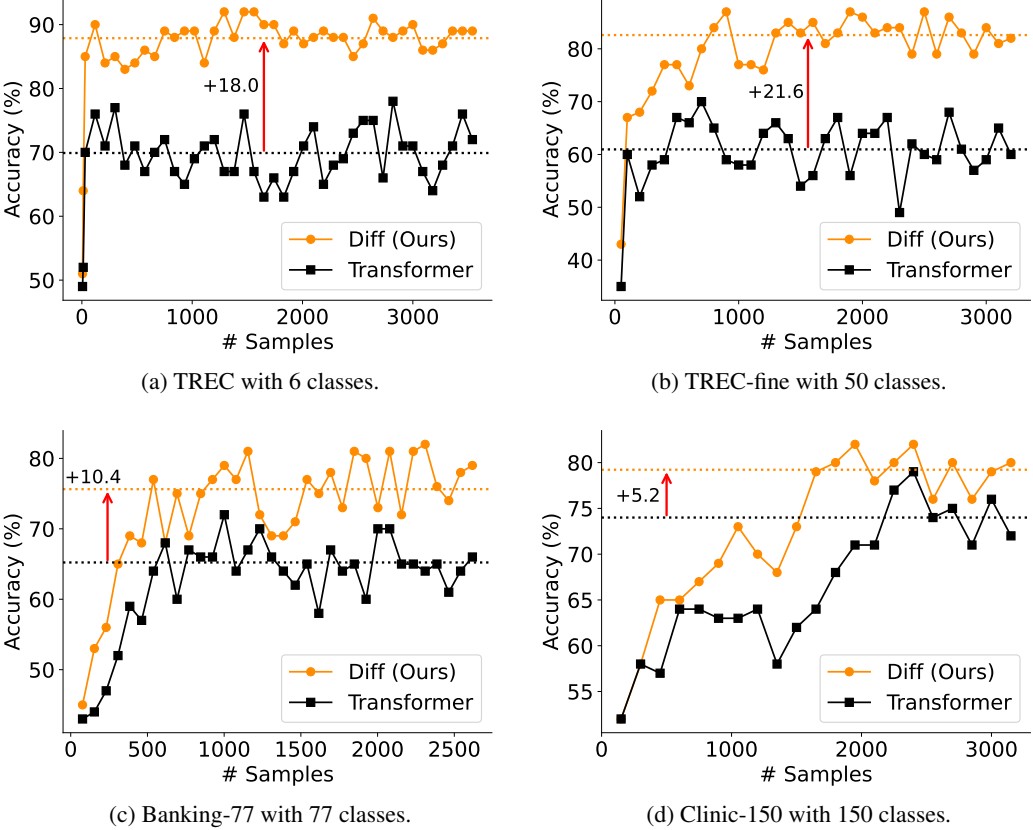

(a) TREC with 6 classes.

(b) TREC-fine with 50 classes.

(c) Banking-77 with 77 classes.

(d) Clinic-150 with 150 classes.

Figure 6: Many-shot in-context learning accuracy on four datasets. Demonstration examples increase from 1-shot until the total length reaches 64K tokens. The dashed lines represent the average accuracy after the performance becomes stable.

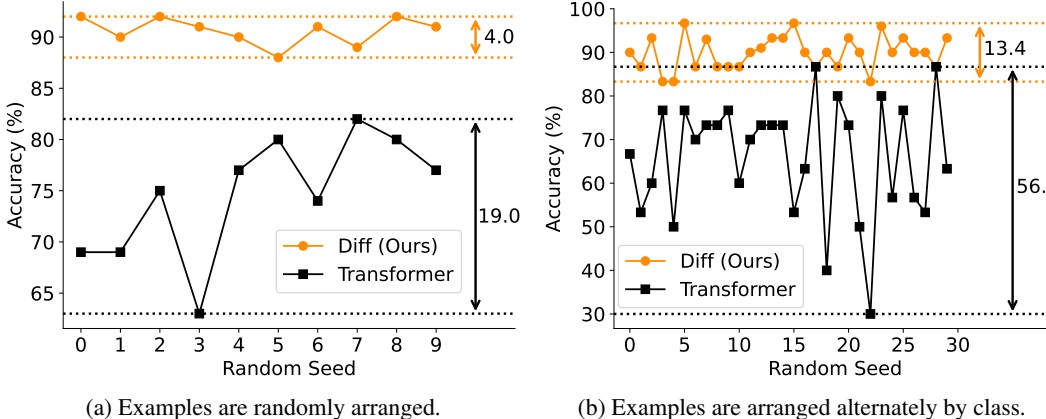

(a) Examples are randomly arranged.

(b) Examples are arranged alternately by class.

Figure 7: Robustness evaluation of in-context learning on the TREC dataset. Accuracy is evaluated with order permutations of demonstration examples by sweeping random seeds. The dash lines represent the margin between the best and worst results. Smaller margin indicates superior robustness. Two prompt formats are examined.

**Robustness of In-Context Learning**   Figure 7 compares the robustness of in-context learning between Transformer and DIFF Transformer. Given the same demonstration examples, we analyze the performance variance with order permutations. Lower variance indicates greater robustness and less risk of catastrophic performance degradation. The evaluation protocol is the same as above. Figure 7 presents the analysis on the TREC dataset. More results are also provided in Appendix F. We evaluate two prompt formats, i.e., examples are randomly arranged (Figure 7a), and alternately arranged by class (Figure 7b). In both settings, DIFF Transformer has much smaller performance variance compared to Transformer. The results indicate that our approach is more robust for in-context learning. In contrast, Transformer tends to be distracted by order permutations (Lu et al., 2022), resulting in a huge margin between the best and worst results.

## 3.6   CONTEXTUAL HALLUCINATION EVALUATION

We evaluate contextual hallucination of the 3B-size language models (described in Appendix B) on text summarization and question answering. Notice that we focus on the cases where the input context contains correct facts, but the model still fails to produce accurate outputs.

We follow the evaluation protocol of (Chuang et al., 2024). We feed the model output along with ground-truth responses to GPT-4o (OpenAI, 2024). Then we ask GPT-4o to make binary judgements on whether the model outputs are accurate and free of hallucinations. Previous studies (Chuang et al., 2024; Ravi et al., 2024) have shown that the above hallucination evaluation protocol has relatively high agreement between GPT-4o judgments and human annotations. The automatic metric is reliable and mirrors the human evaluation. For each dataset, the accuracy is averaged over 100 samples.

**Summarization**   Table 4a presents hallucination evaluation on summarization datasets XSum (Narayan et al., 2018), CNN/DM (See et al., 2017), and MultiNews (Fabbri et al., 2019). The task is to generate summaries for input documents.

| Model | XSum | CNN/DM | MultiNews |
|---|---|---|---|
| Transformer | 0.44 | 0.32 | 0.42 |
| DIFF | **0.53** | **0.41** | **0.61** |

| Model | Qasper | HotpotQA | 2WikiMQA |
|---|---|---|---|
| Transformer | 0.28 | 0.36 | 0.29 |
| DIFF | **0.39** | **0.46** | **0.36** |

(a) Accuracy (i.e., free of hallucinations) on text summarization datasets.

(b) Accuracy (i.e., free of hallucinations) on question answering datasets.

Table 4: Evaluation of contextual hallucination on text summarization and question answering. Higher accuracy indicates less hallucination. We follow Chuang et al. (2024) to employ GPT-4o to make binary judgments, which has relatively high agreement with human annotation.

| Model | Activation Type | Top-1 | Top-2 | Top-3 | Top-10 | Top-100 | Median |
|---|---|---|---|---|---|---|---|
| Transformer | Attention Logits | 318.0 | 308.2 | 304.9 | 284.7 | 251.5 | 5.4 |
| DIFF | Attention Logits | 38.8 | 38.8 | 37.3 | 32.0 | 27.4 | 3.3 |
| Transformer | Hidden States | 3608.6 | 3607.4 | 3603.6 | 3552.1 | 2448.2 | 0.6 |
| DIFF | Hidden States | 1688.2 | 1672.5 | 1672.1 | 1624.3 | 740.9 | 1.2 |

Table 5: Largest activation values in attention logits and hidden states. Top activation values are considered as activation outliers, due to their significantly higher magnitude than the median. DIFF Transformer mitigates outliers compared to Transformer.

**Question Answering**  As shown in Table 4b, we compare the hallucination rate of DIFF Transformer and Transformer on both single- and multi-document question answering. The Qasper (Dasigi et al., 2021) dataset is single-document question answering. In contrast, HotpotQA (Yang et al., 2018) and 2WikiMultihopQA (Ho et al., 2020) are multi-document question answering. The goal is to answer questions about the given context. All evaluation examples are from LongBench (Bai et al., 2023).

Compared with Transformer, our method mitigates contextual hallucination on summarization and question answering. The performance improvement possibly stems from DIFF Transformer's better focus on essential information needed for the task, instead of irrelevant context. This aligns with previous observation (Huang et al., 2024) that one primary reason for contextual hallucination in Transformer is the misallocation of attention scores.

### 3.7 ACTIVATION OUTLIERS ANALYSIS

In large language models, a subset of activations manifests with significantly larger values compared to the majority, a phenomenon commonly called activation outliers (Bondarenko et al., 2024; Sun et al., 2024). The outliers result in difficulties for model quantization during training and inference. We demonstrate that DIFF Transformer can reduce the magnitude of activation outliers, potentially allowing lower bit-widths for quantization.

**Statistics of Largest Activation Values**  Table 5 presents the statistics of activation values collected from Transformer and DIFF Transformer models trained in Appendix B. We analyze two types of activations, including attention logits (i.e., pre-softmax activations), and hidden states (i.e., layer outputs). The statistics are gathered from 0.4M tokens. As shown in Table 5, although the median values are of similar magnitude, DIFF Transformer exhibits much lower top activation values compared to Transformer. The results show that our method produces fewer activation outliers.

**Quantization of Attention Logits**  As shown in Figure 8, we quantize the attention logits to lower bits. We apply dynamic post-training quantization using absmax quantization (Wan et al., 2024). The 16-bit configuration represents the original results without quantization. The models are progressively quantized to 8 bits, 6 bits, and 4 bits. Figure 8 reports the zero-shot performance on HellaSwag (Gao et al., 2023). The other datasets follow a similar trend. DIFF Transformer retains high performance even at reduced bit-widths, ranging from 16 bits to 6 bits. In comparison, Transformer's accuracy significantly drops with 6-bit quantization. The 4-bit DIFF Transformer achieves comparable accuracy as the 6-bit Transformer, and outperforms the 4-bit Transformer by about 25% in accuracy. The results indicate that DIFF Transformer natively mitigates activation outliers in attention scores, providing new opportunities for low-bit FlashAttention (Dao et al., 2022) implementations.

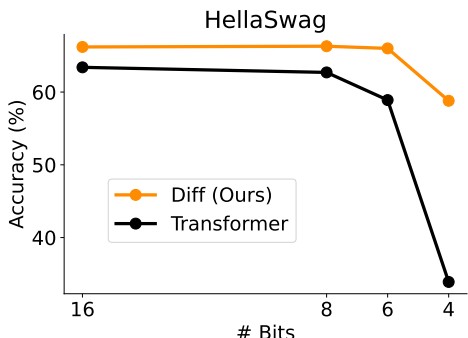

Figure 8: Zero-shot accuracy on the HellaSwag (Gao et al., 2023) dataset. We quantize the attention logits from 16 bits (i.e., unquantized) to 8 bits, 6 bits, and 4 bits.

| Model | #heads | $d$ | GN | Valid. Set↓ | Fine-Grained Slices | |
| | | | | | AR-Hit↓ | Others↓ |
|---|---|---|---|---|---|---|
| Transformer | 16 | 128 | ✗ | 3.087 | 0.898 | 3.272 |
| Transformer | 8 | 256 | ✗ | 3.088 | 0.899 | 3.273 |
| + GroupNorm | 8 | 256 | ✓ | 3.086 | 0.899 | 3.271 |
| DIFF Transformer | 8 | 128 | ✓ | **3.062** | **0.880** | **3.247** |
| − GroupNorm | 8 | 128 | ✗ | 3.122 | 0.911 | 3.309 |
| with $\lambda_{\text{init}} = 0.8$ | 8 | 128 | ✓ | 3.065 | 0.883 | 3.250 |
| with $\lambda_{\text{init}} = 0.5$ | 8 | 128 | ✓ | 3.066 | 0.882 | 3.251 |

Table 6: Ablation studies of 1.4B-size models. We report language modeling loss on the validation set. We also follow Arora et al. (2023) to report fine-grained metrics, where "AR-Hit" evaluates $n$-grams previously seen in the context. "#Heads" is number of heads. "$d$" is head dimension. "GN" indicates whether GroupNorm is used.

## 3.8 ABLATION STUDIES

We conduct ablation studies with 1.4B-size language models. The training setup is the same as the 1.4B model in Section 3.2. The models have $L = 24$ layers, $h = 16$ heads for Transformer, and $h = 8$ heads for DIFF Transformer. The head dimension is $d = 128$. Detailed hyperparameters are described in Appendix E.

Table 6 reports fine-grained loss on the validation set. We follow Zoology (Arora et al., 2023) and divide loss into "*Ar-Hit*" and "*Others*". Specifically, "*Ar-Hit*" considers the last token of an $n$-gram previously seen in the context, which evaluates the associative recall capability. The "*Others*" slice represents the tokens that cannot be recalled from the context or frequent tokens.

As shown in Table 6, we ablate various design choices of DIFF Transformer and present several Transformer variants. Notice that all models have comparable size and training FLOPs for fair comparisons. The first and fourth rows are the default settings for Transformer and DIFF Transformer, respectively, which are directly taken from Figure 3a. Our method outperforms Transformer in terms of both overall and fine-grained loss. As DIFF Transformer halves the number of heads to match model size, the second row shows that the configuration change does not have much impact. We ablate GroupNorm from DIFF Transformer, which degrades performance due to training instability. Because multiple heads tend to have different statistics in our method, GroupNorm plays a key role in normalizing them to similar values. In contrast, comparing the third and first rows, adding GroupNorm to Transformer has negligible effect on performance. The results indicate that the improvements of our method come from the differential attention mechanism, instead of configurations or normalization modules. Moreover, we compare different strategies to initialize $\lambda$. As described in Section 2.1, the default setting uses exponential initialization, i.e., $\lambda_{\text{init}} = 0.8 - 0.6 \times \exp(-0.3 \cdot (l - 1))$, where $l$ is the layer index. The last two rows employ constant initialization with $\lambda_{\text{init}} = 0.8, 0.5$. The minimal change in the validation loss suggests that the models are robust to the choice of $\lambda$ initialization.

## 4 CONCLUSION

In this work, we introduce Differential Transformer (a.k.a. DIFF Transformer), which amplifies attention to the relevant context while canceling noise. Experimental results on language modeling show that DIFF Transformer outperforms Transformer in terms of scaling properties, long-context modeling, key information retrieval, hallucination mitigation, in-context learning, and reduction of activation outliers. The results emphasize the importance of reducing attention noise. Moreover, the differential attention mechanism can be easily implemented with FlashAttention (Dao et al., 2022). The findings position DIFF Transformer as a distinctive and promising foundation architecture for large language models. In the future, we can develop efficient low-bit attention kernels due to the reduced magnitude of activation outliers. As the attention pattern becomes much sparser, we would also like to utilize the property to compress key-value caches.

## ACKNOWLEDGEMENT

We would like to acknowledge Ben Huntley for maintaining the GPU cluster. The long-sequence training utilizes CUBE, which is an internal version of (Lin et al., 2023).

The work is supported in part by the National Key R&D Program of China under Grant 2024YFB4708200 and National Natural Science Foundation of China under Grant U24B20173.

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

## A  IMPLEMENTATION OF DIFFERENTIAL ATTENTION

We present the pseudocode for $\text{DiffAttn}(\cdot)$ and conventional $\text{softmax}$ attention.

```
def Attention(X, W_q, W_k, W_v):        def DiffAttn(X, W_q, W_k, W_v, λ):
    Q = X @ W_q                             Q1, Q2 = split(X @ W_q)
    K = X @ W_k                             K1, K2 = split(X @ W_k)
    V = X @ W_v                             V = X @ W_v
    # Q, K, V: [b, n, d]                    # Qi, Ki: [b, n, d]; V: [b, n, 2d]
    s = 1 / sqrt(d)                         s = 1 / sqrt(d)
    A = Q @ K.transpose(-1, -2) * s         A1 = Q1 @ K1.transpose(-1, -2) * s
                                            A2 = Q2 @ K2.transpose(-1, -2) * s
    return                                  return
      softmax(A) @ V                          (softmax(A1) - λ softmax(A2)) @ V
```

**Implementation with FlashAttention**  Additionally, we provide implementations with FlashAttention (Dao et al., 2022). We categorize the implementations into two types by whether it supports using different dimensions between $Q, K$ and $V$. Specifically, let $\text{FlashDiffAttn\_1}(\cdot)$ denote the package that supports different dimensions (e.g., xformers[1]), and $\text{FlashDiffAttn\_2}(\cdot)$ the package that does not (e.g., flash-attention[2]). We also implement a customized-flash-attention[3] package, which is modified based on the official FlashAttention2 (Dao, 2023), in order to support different dimensions between $Q, K$ and $V$.

The code implementation is available at https://aka.ms/Diff-Transformer.

```
def FlashDiffAttn_1(X, W_q, W_k, W_v, λ):   def FlashDiffAttn_2(X, W_q, W_k, W_v, λ):
    Q1, Q2 = split(X @ W_q)                     Q1, Q2 = split(X @ W_q)
    K1, K2 = split(X @ W_k)                     K1, K2 = split(X @ W_k)
    V = X @ W_v                                 V1, V2 = split(X @ W_v)

    A1 = flash_attn(Q1, K1, V)                  A11 = flash_attn(Q1, K1, V1)
                                                A12 = flash_attn(Q1, K1, V2)
                                                A1 = Concat(A11, A12)
    A2 = flash_attn(Q2, K2, V)                  A21 = flash_attn(Q2, K2, V1)
                                                A22 = flash_attn(Q2, K2, V2)
                                                A2 = Concat(A21, A22)
    return A1 - λ A2                            return A1 - λ A2
```

**Efficiency**  Table 7 compares the throughput between DIFF Transformer and Transformer. For fair comparison, we use the customized-flash-attention implementation mentioned above for both methods. The experiments are conducted with Nvidia H100-80GB GPU cards.

| Model | Model Size | Length | Throughput | |
|---|---|---|---|---|
| | | | **Forward + Backward** | **Forward** |
| Transformer | 3B | 2K | 7247 | 51228 |
| DIFF | 3B | 2K | 6635 $(-9\%)$ | 46811 $(-9\%)$ |
| Transformer | 3B | 4K | 7491 | 48762 |
| DIFF | 3B | 4K | 6718 $(-12\%)$ | 44521 $(-10\%)$ |
| Transformer | 13B | 2K | 998 | 14346 |
| DIFF | 13B | 2K | 942 $(-6\%)$ | 13653 $(-5\%)$ |

Table 7: Throughput is measured with number of tokens per second.

As shown in Table 7, we evaluate the settings with different model size (3B, 13B) and context length (2K, 4K). For 3B models, there are 12 heads for DIFF Transformer and 24 heads for Transformer. For

[1]https://github.com/facebookresearch/xformers
[2]https://github.com/Dao-AILab/flash-attention
[3]https://aka.ms/flash-diff

13B model there are 20 heads for DIFF Transformer and 40 heads for Transformer. All models have the same head dimension $d = 128$. Training efficiency consists of forward and backward. Prefill efficiency only includes forward. Table 7 shows that the throughput results are comparable within an acceptable range. Notice that the `customized-flash-attention` implementation is built on FlashAttention2 (Dao, 2023). With the recent release of FlashAttention3 (Shah et al., 2024), the gap of throughput can be further reduced. More advanced kernel implementation, which is specifically designed for differential attention, can also improve throughput.

## B    LANGUAGE MODELING EVALUATION

Following the same setting as in Section 3.1, we train 3B-size language models on 350B tokens and compare DIFF Transformer with Transformer (Vaswani et al., 2017) in various downstream tasks. We use the augmented Transformer architecture as in LLaMA (Touvron et al., 2023). Specifically, the "Transformer" models include improvements in RMSNorm (Zhang & Sennrich, 2019), SwiGLU (Shazeer, 2020; Ramachandran et al., 2017), and removal of bias.

Table 8 reports the zero-shot and 5-shot results on the LM Eval Harness benchmark (Gao et al., 2023). The results show that DIFF Transformer outperforms Transformer across various tasks in both zero-shot and few-shot settings.

| Model | ARC-C | ARC-E | BoolQ | HellaSwag | OBQA | PIQA | WinoGrande | Avg |
|---|---|---|---|---|---|---|---|---|
| *Training with 350B tokens (Zero-Shot)* | | | | | | | | |
| Transformer-3B | 32.2 | 66.8 | **62.9** | 63.4 | 26.2 | 74.5 | 61.6 | 55.4 |
| DIFF-3B | **33.0** | **68.3** | 60.1 | **66.2** | **27.6** | **75.5** | **62.7** | **56.2** |
| *Training with 350B tokens (5-Shot)* | | | | | | | | |
| Transformer-3B | 34.0 | **69.5** | 65.3 | 63.4 | 25.0 | 75.2 | 62.6 | 56.4 |
| DIFF-3B | **35.0** | **69.5** | **67.2** | **66.9** | **27.6** | **76.1** | **63.8** | **58.0** |

Table 8: Comparison of DIFF Transformer with well-trained Transformer language models on LM Eval Harness (Gao et al., 2023). DIFF Transformer achieves better accuracy in the zero- and few-shot settings.

## C    EVALUATION ON MATHEMATICAL REASONING

We continue training the 3B-size language models that support 64K input length (Section 3.3) with math data to evaluate their o1-style (Jaech et al., 2024) reasoning capability. The training consists of two stages: fine-tuning with synthetic math data, and distilling from DeepSeek-R1 (Guo et al., 2025) to promote o1-style reasoning. We evaluate the models across 8 math benchmarks: GSM-8K (Cobbe et al., 2021), MATH (Hendrycks et al., 2021), SVAMP (Patel et al., 2021), ASDiv (Miao et al., 2020), MAWPS (Koncel-Kedziorski et al., 2016), CARP (Zhang et al., 2023), TABMWP (Lu et al., 2023), and CollegeMath (Tang et al., 2024).

**Math Capability Evaluation**   In the first stage, we train both DIFF Transformer and Transformer for additional 20B tokens on synthetic math data (Li et al., 2024). The learning rate is 8e-5. The batch size is 4M tokens. Hyperparameters are kept the same as in Section 3.3. We evaluate the models every 2B tokens from 6B tokens to 20B tokens and report the average accuracy over 8 math datasets. The output length is restricted to 1K tokens. As shown in Figure 9, DIFF Transformer surpasses Transformer in solving mathematical problems. DIFF Transformer starts to substantially outperform Transformer after 15B tokens, reaching an accuracy gap of 11.3% by the end of 20B tokens. The results demonstrate that DIFF Transformer can learn to solve reasoning tasks more effectively than Transformer.

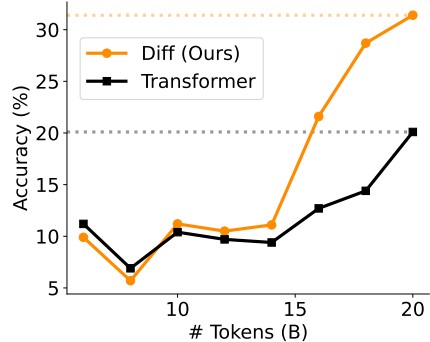

Figure 9: DIFF surpasses Transformer in averaged accuracy over 8 math datasets.

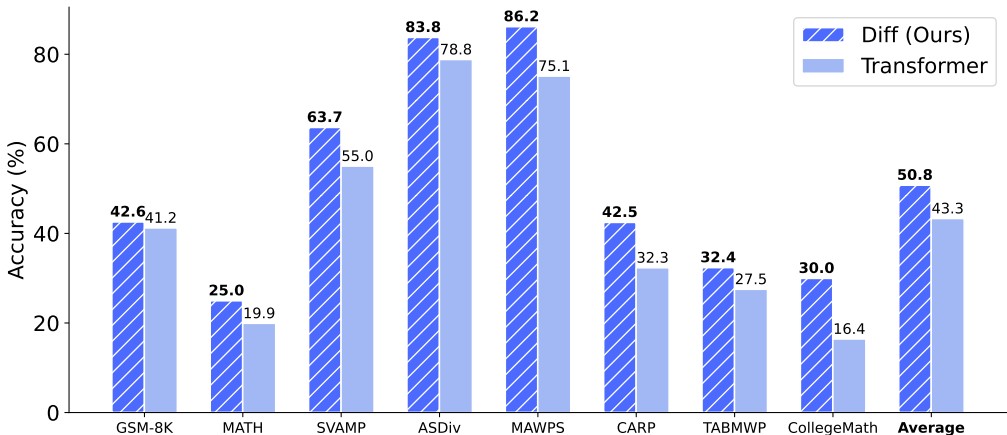

Figure 10: Accuracy on 8 math benchmarks with o1-style reasoning.

**o1-style Reasoning Evaluation**   In the second stage, we further distill the models from OpenThoughts-114K-Math (Open-R1, 2025). The dataset is filtered from OpenThoughts-114K (OpenThoughts, 2025) dataset. It consists of 89K math samples with the average length of 6K tokens. We apply supervised fine-tuning on the dataset to equip the models with o1-style reasoning capability. The learning rate is set to 1e-5. The batch size is 1M tokens. Other hyperparameters are the same as in the first stage. We train both models for 2B tokens and select the best checkpoint for each. The output length is restricted to 16K tokens. As shown in Figure 10, DIFF Transformer outperforms Transformer on all benchmarks with an average accuracy gain of 7.5%. DIFF Transformer generates reasoning process with an average length of 6144 tokens, compared to 6913 for Transformer. The experimental results demonstrate the superior reasoning capability of DIFF Transformer over Transformer. It suggests that differential attention mechanism contributes to the improved performance in mathematical reasoning.

# D  HYPERPARAMETERS FOR SECTION 3.1

Table 9 presents the detailed hyperparameters for the DIFF Transformer-3B models in Section 3.1. For Transformer-3B, the only difference is that there are 24 heads. Notice that both Transformer-3B and DIFF Transformer-3B have similar FLOPs.

| Params | Values |
|---|---|
| Layers | 28 |
| Hidden size | 3072 |
| FFN size | 8192 |
| Vocab size | 100,288 |
| Heads | 12 |
| Adam $\beta$ | (0.9, 0.95) |
| LR | $3.2 \times 10^{-4}$ |
| Batch size | 4M |
| Warmup steps | 1000 |
| Weight decay | 0.1 |
| Dropout | 0.0 |

Table 9: Hyperparamters used for the DIFF Transformer-3B model in Section 3.1.

# E  HYPERPARAMETERS FOR SECTION 3.2

Table 10 reports the hidden dimension, number of layers, and number of heads of DIFF Transformer for different model sizes. For all model sizes of Transformer, we double the number of heads compared with DIFF Transformer to align parameters. The FFN size is $\frac{8}{3} \times d_{\text{model}}$, where $d_{\text{model}}$ is the hidden dimension. The training length is set to 2048. The batch size is set to 0.25M tokens. We use AdamW (Loshchilov & Hutter, 2019) with $\beta_1 = 0.9, \beta_2 = 0.98$. The learning rate is $1.5 \times 10^{-4}$ for 830M to 2.8B sizes, and $7.5 \times 10^{-5}$ for 6.8B to 13.1B sizes. The warmup step is 375 with linear rate decay. The weight decay is set to 0.05. We train the models with 40k steps, i.e., 10B tokens.

| Size | Hidden Dim. | #Layers | #Heads |
|---|---|---|---|
| 830M | 1536 | 24 | 8 |
| 1.4B | 2048 | 24 | 8 |
| 2.8B | 2560 | 32 | 10 |
| 6.8B | 4096 | 32 | 16 |
| 13.1B | 5120 | 40 | 20 |

Table 10: Model size and hyperparameters used for DIFF Transformer in Section 3.2.

# F ROBUSTNESS OF IN-CONTEXT LEARNING

As described in Section 3.5, we evaluate the robustness of in-context learning of Transformer and DIFF Transformer with permutations of the same in-context examples. We evaluate the 3B-size language models that are extended to 64K length (Section 3.3).

Figure 11 provides comparisons on four datasets, with in-context examples randomly arranged. The evaluation protocol is the same as in Section 3.5. The variance in accuracy of DIFF Transformer is consistently lower than that of Transformer, indicating greater robustness of DIFF Transformer for in-context learning.

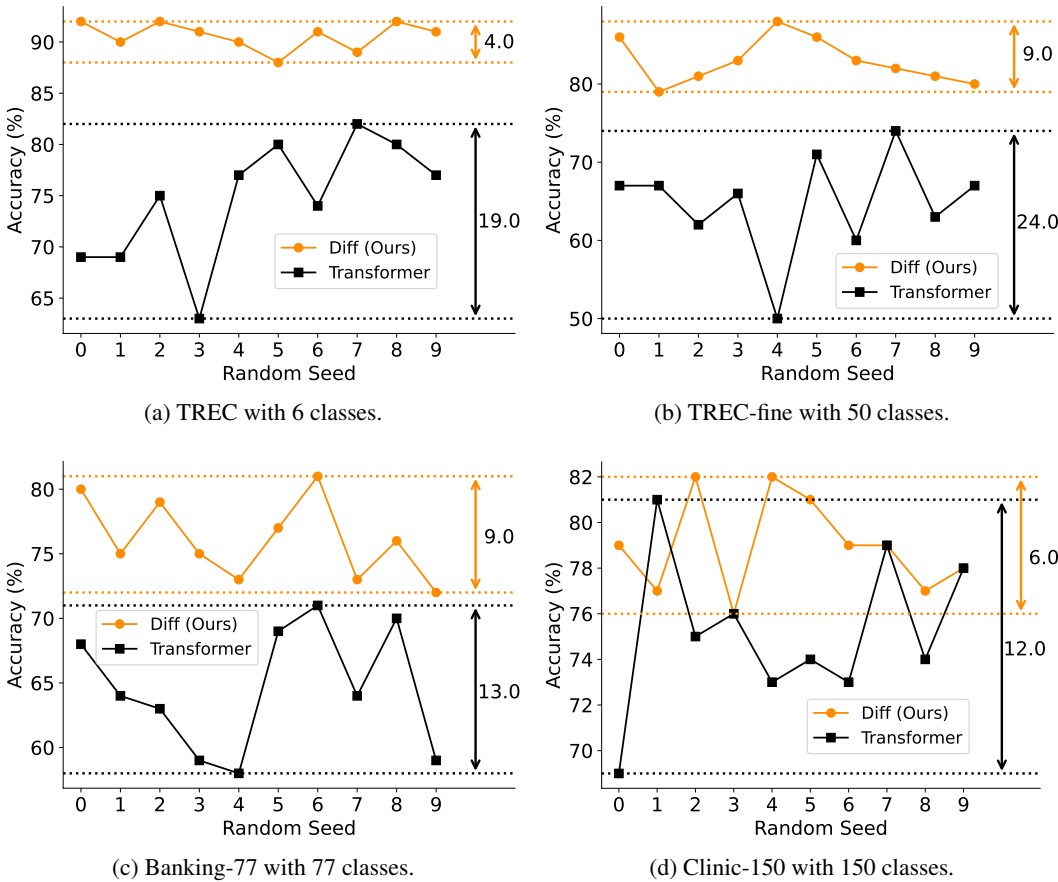

Figure 11: Robustness evaluation of in-context learning on four datasets. Accuracy is evaluated with order permutations of demonstration examples by sweeping random seeds. The dash lines represent the margin between the best and worst results. Demonstration examples are randomly arranged in the prompt.

## G    GRADIENT FLOW OF DIFF TRANSFORMER

We show that the gradient flow in differential attention is similar to that of conventional softmax attention. With this property, the same hyperparameters used in Transformer can be applied directly to the corresponding DIFF Transformer without concerns about training instability.

For differential attention, we select a single head in the proof and expand Equation (1) and Equation (3) as follows. We have $X \in \mathbb{R}^{N \times d_{\text{model}}}$ as the input, $Q_1, Q_2, K_1, K_2 \in \mathbb{R}^{N \times d}$, $V \in \mathbb{R}^{N \times 2d}$, and $O \in \mathbb{R}^{N \times d_{\text{model}}}$ as the output:

$$[Q_1; Q_2] = [XW^{Q_1}; XW^{Q_2}], \quad [K_1; K_2] = [XW^{K_1}; XW^{K_2}], \quad V = XW^V$$

$$A_1 = \text{softmax}(\frac{Q_1 K_1^T}{\sqrt{d}}), \quad A2 = \text{softmax}(\frac{Q_2 K_2^T}{\sqrt{d}}) \tag{6}$$

$$O = \text{GroupNorm}((A_1 - \lambda\, A_2)V)W^O$$

where $W^{Q_1}$, $W^{Q_2}$, $W^{K_1}$, $W^{K_2} \in \mathbb{R}^{d_{\text{model}} \times d}$, $W^V \in \mathbb{R}^{d_{\text{model}} \times 2d}$, $W^O \in \mathbb{R}^{2d \times d_{\text{model}}}$ are parameters, $\lambda$ is a learnable scalar, and GroupNorm has a fixed multiplier as scale: $\gamma = 1 - \lambda_{\text{init}}$. For a token $x$ in $(A_1 - \lambda\, A_2)V$, we have $\dfrac{\partial \text{GN}(x)}{\partial x} = \Theta(\dfrac{\sqrt{2d} \cdot \gamma}{||x||_2}) = \Theta(1)$ as $\dfrac{||x||_2}{\sqrt{2d}} = \Theta(1 - \lambda_{\text{init}})$ at the early training stage. With this formulation and given the gradient of $O$ as $\dfrac{\partial L}{\partial O}$, we formulate gradients of parameters as:

$$\frac{\partial L}{\partial W^O} = \frac{\partial L}{\partial O}\frac{\partial O}{\partial W^O}$$

$$= ((A_1 - \lambda\, A_2)V)^{\mathsf{T}}\frac{\partial L}{\partial O}$$

$$\frac{\partial L}{\partial W^V} = \frac{\partial L}{\partial O}\frac{\partial O}{\partial V}\frac{\partial V}{\partial W^V}$$

$$= X^{\mathsf{T}}(A_1 - \lambda\, A_2)^{\mathsf{T}}\frac{\partial L}{\partial O}(W^O)^{\mathsf{T}}$$

$$\frac{\partial L}{\partial W^{Q_1}} = \frac{\partial L}{\partial O}\frac{\partial O}{\partial A_1}\frac{\partial A_1}{\partial Q_1}\frac{\partial Q_1}{\partial W^{Q_1}}$$

$$= \frac{1}{\sqrt{d}}X^{\mathsf{T}}[A_1 \odot (\frac{\partial L}{\partial O}(W^O)^{\mathsf{T}}V^{\mathsf{T}} - (A_1 \odot (\frac{\partial L}{\partial O}(W^O)^{\mathsf{T}}V^{\mathsf{T}}))J)]K_1$$

$$\frac{\partial L}{\partial W^{Q_2}} = \frac{\partial L}{\partial O}\frac{\partial O}{\partial A_2}\frac{\partial A_2}{\partial Q_2}\frac{\partial Q_2}{\partial W^{Q_2}} \tag{7}$$

$$= \frac{-\lambda}{\sqrt{d}}X^{\mathsf{T}}[A_2 \odot (\frac{\partial L}{\partial O}(W^O)^{\mathsf{T}}V^{\mathsf{T}} - (A_2 \odot (\frac{\partial L}{\partial O}(W^O)^{\mathsf{T}}V^{\mathsf{T}}))J)]K_2$$

$$\frac{\partial L}{\partial W^{K_1}} = \frac{\partial L}{\partial O}\frac{\partial O}{\partial A_1}\frac{\partial A_1}{\partial K_1}\frac{\partial K_1}{\partial W^{K_1}}$$

$$= \frac{1}{\sqrt{d}}X^{\mathsf{T}}[A_1 \odot (\frac{\partial L}{\partial O}(W^O)^{\mathsf{T}}V^{\mathsf{T}} - (A_1 \odot (\frac{\partial L}{\partial O}(W^O)^{\mathsf{T}}V^{\mathsf{T}}))J)]^{\mathsf{T}}Q_1$$

$$\frac{\partial L}{\partial W^{K_2}} = \frac{\partial L}{\partial O}\frac{\partial O}{\partial A_2}\frac{\partial A_2}{\partial K_2}\frac{\partial K_2}{\partial W^{K_2}}$$

$$= \frac{-\lambda}{\sqrt{d}}X^{\mathsf{T}}[A_2 \odot (\frac{\partial L}{\partial O}(W^O)^{\mathsf{T}}V^{\mathsf{T}} - (A_2 \odot (\frac{\partial L}{\partial O}(W^O)^{\mathsf{T}}V^{\mathsf{T}}))J)]^{\mathsf{T}}Q_2$$

where $J \in \mathbb{R}^{N \times N}$ is a all-one matrix.

As a comparison, we reformulate conventional softmax attention. For attention with $2d$ dimension, we have $X \in \mathbb{R}^{N \times d_{\text{model}}}$ as the input, $Q_1, Q_2, K_1, K_2 \in \mathbb{R}^{N \times d}$, $V \in \mathbb{R}^{N \times 2d}$, and $O \in \mathbb{R}^{N \times d_{\text{model}}}$ as

the output:

$$[Q_1; Q_2] = [XW^{Q_1}; XW^{Q_2}], \quad [K_1; K_2] = [XW^{K_1}; XW^{K_2}], \quad V = XW^V$$

$$A = \mathrm{softmax}(\frac{Q_1 K_1^T + Q_2 K_2^T}{\sqrt{2d}}) \tag{8}$$

$$O = (AV)W^O$$

where $W^{Q_1}, W^{Q_2}, W^{K_1}, W^{K_2} \in \mathbb{R}^{d_{\mathrm{model}} \times d}, W^V \in \mathbb{R}^{d_{\mathrm{model}} \times 2d}, W^O \in \mathbb{R}^{2d \times d_{\mathrm{model}}}$ are parameters. Denote the gradient of $O$ as $\frac{\partial L}{\partial O}$, we formulate gradients of parameters via:

$$
\begin{aligned}
\frac{\partial L}{\partial W^O} &= \frac{\partial L}{\partial O}\frac{\partial O}{\partial W^O} \\
&= (AV)^\intercal \frac{\partial L}{\partial O} \\
\frac{\partial L}{\partial W^V} &= \frac{\partial L}{\partial O}\frac{\partial O}{\partial V}\frac{\partial V}{\partial W^V} \\
&= X^\intercal A^\intercal \frac{\partial L}{\partial O}(W^O)^\intercal \\
\frac{\partial L}{\partial W^{Q_1}} &= \frac{\partial L}{\partial O}\frac{\partial O}{\partial A}\frac{\partial A}{\partial Q_1}\frac{\partial Q_1}{\partial W^{Q_1}} \\
&= \frac{1}{\sqrt{2d}}X^\intercal[A \odot (\frac{\partial L}{\partial O}(W^O)^\intercal V^\intercal - (A \odot (\frac{\partial L}{\partial O}(W^O)^\intercal V^\intercal))J]K_1 \\
\frac{\partial L}{\partial W^{Q_2}} &= \frac{\partial L}{\partial O}\frac{\partial O}{\partial A}\frac{\partial A}{\partial Q_2}\frac{\partial Q_2}{\partial W^{Q_2}} \\
&= \frac{1}{\sqrt{2d}}X^\intercal[A \odot (\frac{\partial L}{\partial O}(W^O)^\intercal V^\intercal - (A \odot (\frac{\partial L}{\partial O}(W^O)^\intercal V^\intercal))J]K_2 \\
\frac{\partial L}{\partial W^{K_1}} &= \frac{\partial L}{\partial O}\frac{\partial O}{\partial A}\frac{\partial A}{\partial K_1}\frac{\partial K_1}{\partial W^{K_1}} \\
&= \frac{1}{\sqrt{2d}}X^\intercal[A \odot (\frac{\partial L}{\partial O}(W^O)^\intercal V^\intercal - (A \odot (\frac{\partial L}{\partial O}(W^O)^\intercal V^\intercal))J]^\intercal Q_1 \\
\frac{\partial L}{\partial W^{K_2}} &= \frac{\partial L}{\partial O}\frac{\partial O}{\partial A}\frac{\partial A}{\partial K_2}\frac{\partial K_2}{\partial W^{K_2}} \\
&= \frac{1}{\sqrt{2d}}X^\intercal[A \odot (\frac{\partial L}{\partial O}(W^O)^\intercal V^\intercal - (A \odot (\frac{\partial L}{\partial O}(W^O)^\intercal V^\intercal))J]^\intercal Q_2
\end{aligned}
\tag{9}
$$

With the property of $\mathrm{softmax}$, we have $A \overset{\Theta}{=} A_1 \overset{\Theta}{=} A_2 \overset{\Theta}{=} A_1 - \lambda A_2$, considering gradient magnitude. Therefore, the gradients of the corresponding parameters of attention and differential attention are equivalent in magnitude, differing by some constant factors, as shown in Equation (7) and Equation (9). When using an optimizer that is invariant to gradient magnitude, such as AdamW (Loshchilov & Hutter, 2019), parameter updates in DIFF Transformer are similar to those of Transformer. This allows us to reuse Transformer hyperparameters without risking training instability.

