# OpenReview forum: "Differential Transformer"
_ICLR.cc/2025/Conference — ICLR 2025 Oral_

### Official Review · Reviewer_txCS · 2024-11-04

**Soundness:** 3
**Presentation:** 3
**Contribution:** 3
**Rating:** 8
**Confidence:** 4

**Summary:**

The paper introduces a new Transformer variant, called Diff Transformer. The key idea is in the design of attention mechanism, where Diff Transformer uses two softmax attention functions to cancel out potential attention noises. Through a suite of empirical studies, Diff Transformer show promising performances compared to standard Transformer architecture.

**Strengths:**

- The paper introduces a simple architecture tweak to mitigate the observations that Transformer models tend to incorrectly allocate excessive attentions to irrelevant contexts.
- The paper demonstrates promising empirical results of the proposed architecture through a decent suite of evaluations, ranging from language modeling capability that fits well with scaling law, in-context learning, to improving long-context capability and mitigating contextual hallucination.
- I appreciate that the authors include a variety of downstream tasks to showcase how addressing attention noises in Transformer models can lead to performance improvements.

**Weaknesses:**

- Related work can be discussed more thoroughly in the paper. For example, [1] discusses that Transformer models tend to mis-allocate attention to irrelevant contexts potentially biased by their position within the context [1], and [2] also shows LLMs can be easily distracted by irrelevant contexts.
- While the proposed Diff Transformer shows promising performances, they have to be trained from scratch which can be computationally expensive compared to post-training approaches that mitigate attention noises in standard Transformer models, such as the calibration technique used in [1], and improved decoding approaches used in [3] and [4]. Currently, vanilla Transformer is the only baseline considered in the paper. However, I would appreciate if the authors can show how Diff Transformer compares to other more lightweight techniques that fix Transformer's attention noises.
- Apart from pretraineing Diff Transformer, there is no further instruction tuning experiments included in the current paper. I would be interested in seeing whether instruction tuned Diff Transformer also shows better performances than standard instruction tuned Transformers.


[1] Found in the Middle: Calibrating Positional Attention Bias Improves Long Context Utilization. Hsieh et al. 2024.

[2] Large Language Models Can Be Easily Distracted by Irrelevant Context. Shi et al. 2023.

[3] Trusting Your Evidence: Hallucinate Less with Context-aware Decoding. Shi et al. 2023.

[4] Lookback Lens: Detecting and Mitigating Contextual Hallucinations in Large Language Models Using Only Attention Maps. Chuang et al. 2024.

**Questions:**

- Can the authors elaborate further on why we need a fixed multiplier $(1 − \lambda_{init}) $ in Diff Transformer (Eq. 3)?
- In section 3.7, Is absmax quantization the suitable approach to adopt in this context? Why not consider quantization approaches that are robust to activation outliers, such as LLM.int8()?

**Details Of Ethics Concerns:**

No immediate ethics concerns.

---

> ### Author Response · Authors · 2024-11-19
> **Title: Response to Reviewer txCS (1/2): Response to the Weaknesses Part**
>
> We appreciate the reviewer finds our approach promising and experiments solid. Here are our responses:
>
> **Weaknesses part:**
>
> > 1\. Related work can be discussed more thoroughly in the paper. For example, [1] discusses that Transformer models tend to mis-allocate attention to irrelevant contexts potentially biased by their position within the context, and [2] also shows LLMs can be easily distracted by irrelevant contexts.
>
> We appreciate the compelling related works that the reviewer mentions. In [1], the authors identify that Transformer exhibits an U-shaped attention bias. It tends to misallocate attention to irrelevant contexts at the beginning and end of the sequence. Similarly, [2] investigates the distractibility of Transformer models and finds that performance significantly drops when irrelevant information is introduced into the context. Our observations align with the conclusions of both studies, confirming that Transformer is susceptible to distraction from irrelevant information, and can misallocate attention scores to these terms.
>
> We will add this to discussion section when there are extra pages available.
>
>
>
> > 2\. While the proposed DIFF Transformer shows promising performances, they have to be trained from scratch which can be computationally expensive compared to post-training approaches that mitigate attention noises in standard Transformer models, such as the calibration technique used in [1], and improved decoding approaches used in [3] and [4]. Currently, vanilla Transformer is the only baseline considered in the paper. However, I would appreciate if the authors can show how DIFF Transformer compares to other more lightweight techniques that fix Transformer's attention noises.
>
> [1] mitigates the U-shaped attention bias through a calibration mechanism. [3] utilizes a contrastive method in output distribution, named as context-aware decoding, to improve the faithfulness of LLM. [4] proposes to use the ratio of attention weights as a hallucination detection model and guide decoding with it. These are all effective works that identify problems in LLM similar to attention noise, and mitigate it with lightweight techniques.
>
> In DIFF Transformer, we focus on solving the attention noise problem fundamentally. Training models from scratch helps to fully unleash the potential of the proposed approach. By training from scratch, we can comprehensively enhance DIFF Transformer's capabilities so as to thoroughly validate the benefits of canceling attention noise. Moreover, it is also feasible to apply DIFF approach to pretrained LLMs with post-training methods. It is an interesting and promising topic to explore in the future.
>
> We will add this to discussion section when there are extra pages available.
>
>
>
> [1] Found in the Middle: Calibrating Positional Attention Bias Improves Long Context Utilization. Hsieh et al. 2024.
>
> [2] Large Language Models Can Be Easily Distracted by Irrelevant Context. Shi et al. 2023.
>
> [3] Trusting Your Evidence: Hallucinate Less with Context-aware Decoding. Shi et al. 2023.
>
> [4] Lookback Lens: Detecting and Mitigating Contextual Hallucinations in Large Language Models Using Only Attention Maps. Chuang et al. 2024.
>
> > 3\. Apart from pretraining DIFF Transformer, there is no further instruction tuning experiments included in the current paper. I would be interested in seeing whether instruction tuned DIFF Transformer also shows better performances than standard instruction tuned Transformers.
>
> We believe DIFF Transformer is also effective in instruction tuning experiments. Take hallucination evaluation as an example: in addition to question answering and text summarization, multi-turn dialogue is also an important scenario where dialogue history often districts generation of the next response in instruction-tuned LLMs. By mitigating contextual hallucination, DIFF Transformer also has the potential to perform well in this task.
>
> We are interested in exploring this topic further in the future. We will include updates in the revised version.

---

> ### Author Response · Authors · 2024-11-19
> **Title: Response to Reviewer txCS (2/2): Response to the Questions Part**
>
> **Questions part:**
>
> > 1\. Can the authors elaborate further on why we need a fixed multiplier $(1-\lambda_{\text{init}})$ in DIFF Transformer Equation (3)?
>
> The related part in Equation (3) is:
>
> $$
> \\overline{{\\mathrm{head} _ i}} = (1 - \\lambda _ {\\text{init}}) \\cdot \\operatorname{LN}( \\mathrm{head} _ i ),~ i \in [1, h]
> $$
>
> The multiplier $(1 - \lambda_{\text{init}})$ can be considered as the initialization of the scale parameter of $\operatorname{LN}$, designed to align the gradients with Transformer. At the early training stage,  with differential attention defined in Equation (1), for a token $x$ in $\mathrm{head}_i$ we have:
>
> $$
> \\cfrac{||x|| _ 2}{\\sqrt{d}}=\\Theta(1-\\lambda _ {\\text{init}})
> $$
>
> Consider the gradient of $\operatorname{LN}$:
>
> $$
> (1 - \\lambda _ {\\text{init}}) \\cdot \\cfrac{\\partial \\text{LN}(x)}{\\partial x} = \\Theta((1 - \\lambda _ {\\text{init}}) \\cdot \\cfrac{1}{||x|| _ 2 \\ / \\sqrt{d}}) = \\Theta(\\cfrac{1 - \\lambda _ {\\text{init}}}{1-\\lambda _ {\\text{init}}}) = \\Theta(1)
> $$
>
> With the multiplier $(1 - \lambda_{\text{init}})$, we counteract the gradient factor $\frac{1}{1-\lambda_{\text{init}}} > 1$, better aligning gradients of Diff Transformer with Transformer to avoid hyperparameter adjustments.
>
> The detailed gradient derivation can be found in Appendix E.
>
> > 2\. In section 3.7, Is absmax quantization the suitable approach to adopt in this context? Why not consider quantization approaches that are robust to activation outliers, such as LLM.int8()?
>
> Recent research suggests that activation outliers are closed related to attention mechanism both for attention logits [1] and hidden states [2]. In section 3.7, we aim to investigate whether activation outliers can be reduced natively through attention design. To illustrate the benefits of reducing outliers, we primarily employ absmax quantization. Advanced quantization techniques can also be applied to DIFF Transformer, and the improvement over Transfomer still exists as activation outliers are reduced. We are also exploring quantization-aware training with DIFF Transformer, which shows promising preliminary results.
>
> [1] Quantizable transformers: Removing outliers by helping attention heads do nothing. Bondarenko et al. 2023.
>
> [2] Massive activations in large language models. Sun et al. 2024.

---

> ### Comment · Reviewer_txCS · 2024-11-28
> **Post rebuttal comments**
>
> Thank you to the authors for the response. I hope the authors would include the discussions here into the revision, and I look forward to follow-up results on applying DIFF in a post-pretraining/instruction tuning paradigm.

---

> > ### Author Response · Authors · 2024-11-28
> >
> > Thank you for your recognition of our work. The discussions here will be taken into consideration in the revised version.

---

### Official Review · Reviewer_cGmd · 2024-11-04

**Soundness:** 3
**Presentation:** 3
**Contribution:** 3
**Rating:** 8
**Confidence:** 4

**Summary:**

This paper proposes a new model architecture called Differential Transformer (DIFF Transformer), which reduces noise and more accurately focuses on relevant context through a differential attention mechanism. In a series of language modeling experiments, DIFF Transformer outperforms the standard Transformer across various tasks and model sizes. The paper demonstrates the superiority of this method in long-context modeling, key information retrieval, hallucination mitigation, in-context learning, and reduction of activation outliers.

**Strengths:**

1.The proposed DIFF Transformer introduces an innovative differential attention mechanism that reduces attention noise through the difference between two independent softmax mappings. This approach performs exceptionally well in long-context processing, making a significant improvement for long-text tasks.
2.DIFF Transformer demonstrates strong performance in language modeling, in-context learning, and multi-needle retrieval tasks, particularly outperforming traditional Transformers in long-context scenarios. It greatly enhances the accuracy and robustness in practical applications like question answering and text summarization, showing broad potential for various use cases.
3.The paper provides extensive experimental validation, confirming DIFF Transformer’s stability across different tasks and model parameter settings. The hyperparameter tuning is thorough, ensuring both robust performance and reliable results.
4.The paper is well-organized, with clear illustrations and analogies that present the differential attention mechanism effectively. Detailed experimental steps make it easy to understand the innovative contributions.

**Weaknesses:**

While the differential attention mechanism brings notable performance improvements, it also adds computational complexity. Efficiency tests indicate that DIFF Transformer’s throughput, particularly with multi-head normalization and dual softmax calculations, is slightly lower than that of traditional Transformers (by about 5%-12%). Although this impact is relatively minor in the experiments, further exploration of computational efficiency would be valuable if scaling to large-scale applications.

**Questions:**

I don't have questions.

---

> ### Author Response · Authors · 2024-11-19
> **Title: Response to Reviewer cGmd**
>
> We are glad the reviewer finds our approach innovative and promising. We also appreciate the comment regarding computational efficiency. We list some possible ways to further improve effiency:
>
> - **Impact of Model Size:** As the model size increases, the computation occupied by FFN grows. The gap of throughput of DIFF Transformer will be reduced in larger models, which can also be observed in Table 7.
>
> - **Advanced Kernels:** Efficiency of DIFF Transformer can be further improved by implementing with most advanced FlashAttention techniques such as FlashAttention3 [1] or with kernel implementation designed for differential attention.
>
> [1] Flashattention-3: Fast and accurate attention with asynchrony and low-precision. Shah et al. 2024.

---

### Official Review · Reviewer_QNSJ · 2024-11-06

**Soundness:** 3
**Presentation:** 4
**Contribution:** 3
**Rating:** 8
**Confidence:** 4

**Summary:**

The paper introduces Differential Transformer, a novel architecture for LLMs that enhances attention to relevant context while canceling out noise. The core innovation is a differential attention mechanism that calculates attention scores as the difference between two softmax attention maps, promoting sparse attention patterns and reducing attention noise. The paper demonstrates through extensive experiments that the DIFF Transformer outperforms the standard Transformer in various aspects, including scaling model size, training tokens, long-context modeling, key information retrieval, hallucination mitigation, in-context learning, and reduction of activation outliers.

**Strengths:**

1. this is a solid and well-written paper.
2. while the technique itself is not complex, as far as I know, this is the first work to propose a new architecture using differential attention. 3. the experiments and conclusions in this paper are thorough and insightful.

**Weaknesses:**

1. the authors mention "promoting the emergence of sparse attention patterns" multiple times on Lines 13, 160, and 539, but do not provide statistics and quantification of the sparsity of attention distribution between DIFF Transformer and the general Transformer.
2. The authors did not discuss or conduct experimental comparisons with work related to the sparse attention.
For example:
> Efficient Content-Based Sparse Attention with Routing Transformers
>
> Generating Long Sequences with Sparse Transformers.
3. It would be even better if the effectiveness of the DIFF Transformer could be validated on image or speech modalities.
4. The absence of a *Related Work* section limits the paper's ability to contextualize its contributions within the broader field.

**Questions:**

1. Do both attention calculations in differential attention use RoPE positional encoding?
2. Similar to multi-query attention, can $K_1$, $K_2$, or $Q_1$, $Q_2$ share parameters?
3. Is there a significant difference in sparsity between DiffAttn and regular attention? Does the first term of DiffAttn resemble the pattern of standard attention (implying that the second negative attention term serves to cancel out noise)?
4. What is the calculation formula for Table 3? (Specifically, what is the exact normalization operation?)
5. Please discuss the relationship between differential attention and sparse attention.

---

> ### Author Response · Authors · 2024-11-19
> **Title: Response to Reviewer QNSJ (1/2): Response to the Weaknesses Part**
>
> We appreciate that the reviewer finds our paper solid and insightful. Here are our responses.
>
> **Weaknesses Part:**
>
> > 1\. The authors mention "promoting the emergence of sparse attention patterns" multiple times, but do not provide statistics and quantification of the sparsity of attention distribution between DIFF Transformer and the general Transformer.
>
> We provide statistics of attention scores to demonstrate the sparse attention pattern of DIFF Transformer in Table 3. The statistics are collected from the key information retrieval task. "Attention to Answer" denotes the sum of attention scores that the generated tokens allocate to the answer span. "Attention Noise" denotes the sum of scores that the generated tokens allocate to the irrelevant noise context.
>
> Compared with Transformer, the generated tokens of DIFF Transformer allocate much higher scores to the answer span, and lower scores to the noise context. Further given that the answer span only includes a few tokens, and the attention scores on the noise context (about 4K tokens) are almost uniformly distributed, the statistics can prove that DIFF Transformer attends to key information more sparsely.
>
> It would be better to clearly point out the relationship of this table to the emergence of sparse attention pattern. We will add this in the revised version.
>
>
> > 2\. The authors did not discuss or conduct experimental comparisons with work related to the sparse attention.
> >
> > 4\. The absence of a Related Work section limits the paper's ability to contextualize its contributions within the broader field.
>
> We appreciate the reviewer’s suggestions for the related work. We compare and discuss them in the response to the fifth question.
>
>
> > 3\. It would be even better if the effectiveness of the DIFF Transformer could be validated on image or speech modalities.
>
> We are conducting experiments on more modalities with DIFF Transformer these days, and there are some positive results. We will add results of more modalities in the revised version.

---

> ### Author Response · Authors · 2024-11-19
> **Title: Response to Reviewer QNSJ (2/2): Response to the Questions Part**
>
> **Questions Part:**
>
> > 1\. Do both attention calculations in differential attention use RoPE positional encoding?
>
> Yes.
>
> >  2\. Similar to multi-query attention, can $K_1$, $K_2$, or $Q_1$, $Q_2$ share parameters?
>
> Yes. We conduct experiments on models with 300M parameters. The validation loss of Transformer is 4.406. For a full-parameter DIFF Transformer, the loss is 4.343. For a DIFF Transformer with $Q_1=Q_2$, the loss is 4.347. For a DIFF Transformer with $K_1=K_2$, the loss is 4.346. The performance indicates it is possible to share $K_1$, $K_2$, or $Q_1$, $Q_2$ in DIFF Transformer.
>
> > 3\. Is there a significant difference in sparsity between DiffAttn and regular attention? Does the first term of DiffAttn resemble the pattern of standard attention (implying that the second negative attention term serves to cancel out noise)?
>
> Yes. We demonstrate the sparse pattern in Figure 1 and Table 3. We further explain it in the response to the first question in the Weaknesses Part above.
>
> We visualized two attention maps.  The sum of the second attention term equals to $\lambda$. For most layers, $\lambda < 1$. Under this circumstance, we find on noise context, the second term allocates almost same scores as the first term, while on key information, the second term allocates less scores than the first term. Therefore after subtraction, the noise is canceled out, and the key information is kept. For a few layers with $\lambda > 1$, the role of the two terms exchanges, and the conclusion is the same.
>
> >  4\. What is the calculation formula for Table 3?
>
> Firstly, we define the token set that contains all key information tokens as $A$, set contains all noise context tokens as $N$, and set contains the generated tokens as $G$.
>
> For conventional attention, we define attention score, i.e. the output of $\operatorname{softmax}$ function, as $a_{ij}$, which means the score that the $i$-th token allocates to the $j$-th token, $1 \le j \le i$.
>
> "Attention to Answer" is calculated with:
> $$
> \\text{Score} _ {\\text{ans}} = \\cfrac{1}{|G|} \\sum _ {i \\in G} \\cfrac{\\sum _ {j \\in A} \\ a_{ij}}{\\sum _ {j=1}^i \\ a_{ij}} = \\cfrac{1}{|G|} \\sum _ {i \\in G} \\cfrac{\\sum _ {j \\in A} \\ a_{ij}}{1}
> $$
>
> "Attention Noise" is calculated with:
> $$
> \\text{Score} _ {\\text{noise}} = \\cfrac{1}{|G|} \\sum _ {i \\in G} \\cfrac{\\sum _ {j \\in N} \\ a _ {ij}}{\\sum _ {j=1}^i \\ a _ {ij}} = \\cfrac{1}{|G|} \\sum _ {i \\in G} \\cfrac{\\sum _ {j \\in N} \\ a _ {ij}}{1}
> $$
>
> For differential attention, we define attention scores of the first term as $a_{ij}^{(1)}$ and scores of the second term as $a_{ij}^{(2)}$.
>
> "Attention to Answer" is calculated with:
> $$
> \\text{Score} _ {\\text{ans}} = \\cfrac{1}{|G|} \\sum _ {i \\in G} \\cfrac{\\sum _ {j \\in A} \\ a _ {ij}^{(1)} - \\lambda a _ {ij}^{(2)}}{\\sum _ {j=1}^i \\ a _ {ij}^{(1)} - \\lambda a _ {ij}^{(2)}} = \\cfrac{1}{|G|} \\sum _ {i \\in G} \\cfrac{\\sum _ {j \\in A} \\ a _ {ij}^{(1)} - \\lambda a _ {ij}^{(2)}}{1-\\lambda}
> $$
>
> "Attention Noise" is calculated with:
> $$
> \\text{Score} _ {\\text{noise}} = \\cfrac{1}{|G|} \\sum _ {i \\in G} \\cfrac{\\sum _ {j \\in N} \\ a _ {ij}^{(1)} - \\lambda a _ {ij}^{(2)}}{\\sum _ {j=1}^i \\ a _ {ij}^{(1)} - \\lambda a _ {ij}^{(2)}} = \\cfrac{1}{|G|} \\sum _ {i \\in G} \\cfrac{\\sum _ {j \\in N} \\ a _ {ij}^{(1)} - \\lambda a _ {ij}^{(2)}}{1-\\lambda}
> $$
>
> In the DIFF Transformer model we report, $\lambda \in (0,1)$ stands for all layers, so $1-\lambda$ is always positive.
>
> The scores are then averaged across all layers and heads to obtain the final scores in Table 3. The calculation formula ensures all reported scores are within range $[0,1]$.
>
> > 5\. Please discuss the relationship between differential attention and sparse attention.
>
> The mentioned works are both influential paper. Routing Transformer [1] assigns queries and keys to clusters and performs attention only within each cluster. The complexity of attention is reduced to $\mathcal{O}(n^{1.5})$ with the proposed method. Sparse Transformer [2] uses sparse factorizations of the attention matrix to reduce the complexity of attention to $\mathcal{O}(n^{1.5})$. Both works create sparsity by restricting attention to a subset of tokens to accelerate attention. The performance of such methods is usually worse than their full attention counterparts due to the discarded tokens. It is also difficult for them to perform well in tasks such as key information retrieval.
>
> In comparison, DIFF Transformer aims to natively create sparsity in full attention. DIFF Transformer can attend to important context accurately and sparsely without discarding any tokens, enabling it to outperm full attention Transformer in language modeling and tasks such as key information retrieval.
>
> We will add this to discussion section when there are extra pages available.
>
> [1] Efficient Content-Based Sparse Attention with Routing Transformers. Roy et al. 2021.
>
> [2] Generating Long Sequences with Sparse Transformers. Child et al. 2019.

---

> ### Comment · Reviewer_QNSJ · 2024-11-21
> **Official Comment by Reviewer QNSJ**
>
> Thank you for your detailed response. I have no further questions. I will keep my very positive score.
>
> Moreover, I would like to suggest modifying "promoting the emergence of sparse attention patterns" to "promoting the emergence of concentrated attention patterns". The term "sparse" implies that some attention scores are zero (which benefits sparse matrix operations), the "concentrated attention" seems more appropriate to describe the effect of differential attention.

---

> > ### Author Response · Authors · 2024-11-21
> >
> > Thank you for your recognition of our work and valuable suggestion. The suggested improvements will be taken into consideration in the revised version.

---

### Official Review · Reviewer_hbKx · 2024-11-10

**Soundness:** 3
**Presentation:** 4
**Contribution:** 3
**Rating:** 8
**Confidence:** 4

**Summary:**

The paper aims to mitigate the problem in standard softmax attention to over-allocate attention weights to irrelevant context. The proposed Diff Transformer uses the difference between two separate softmax attention scores to cancel noise assigned to irrelevant contextual tokens. Concretely, Diff Transformer first partition the query and key vectors into two groups to compute separate softmax attention scores. The final attention score is the subtraction of the two groups. One learnable parameter $\lambda$ is introduced to balance the two groups of attention scores.

Experiments were conducted on a 3B-parameter model trained on 350B data tokens. Diff Transformer outperforms Transformer on training loss and downstream benchmarks. Ablation studies investigate the importance of the RMSNorm on each attention head.

**Strengths:**

1. The proposed modification of attention in Diff Transformer is well-motivated

2. The experimental results are strong, with large-scale experiments up to 3B-parameter models and 350B data tokens.

3. The pre-trained model was evaluated on multiple benchmarks, and also on long-context evaluation, retrieval-oriented tasks, many-shot in-context learning and hallucination evaluation.

4. The paper is well-written, easy to follow.

**Weaknesses:**

Some design motivation in the model is still not clear:

1. Why the learnable $\lambda$ is re-parameterized in Eq (2)?

2. Why in Eq (3) there is a term $(1 - \lambda_{init}$ for each head?

3. Why the RMSNorm for each head is so important for Diff Transformer stability? The explainable in section 3.8 is unconvincing to me.

**Questions:**

NA

---

> ### Author Response · Authors · 2024-11-19
> **Title: Response to Reviewer hbKx**
>
> We are glad the reviewer finds our idea well-motivated and our experimental results strong. We respond to the reviewer’s questions below.
>
> > 1\. Why the learnable $\lambda$ is re-parameterized in Equation (2)?
>
> $\lambda$ is multiplied to $\operatorname{softmax}$, where $\operatorname{softmax}(\mathbf{q}, \mathbf{k}) = \frac{\exp(\mathbf{q} \cdot \mathbf{k})}{\sum \exp(\mathbf{q} \cdot \mathbf{k})}$. Parameters in $\lambda$ learn with the same learning rate as other parameters in the model, therefore $\lambda$ should take a similar formulation as  $\operatorname{softmax}$ to synchronize the learning dynamics.
>
> Consider the re-parameterization  $\lambda = \exp(\mathbf{\lambda_{q}} \cdot \mathbf{\lambda_{k}}),~ \mathbf{\lambda_{q}}, \mathbf{\lambda_{k}} \in \mathbb{R}^d $, we then have $\lambda \cdot \operatorname{softmax}(\mathbf{q}, \mathbf{k}) = \frac{\exp(\mathbf{q} \cdot \mathbf{k} + \mathbf{\lambda_{q}} \cdot \mathbf{\lambda_{k}})}{\sum \exp(\mathbf{q} \cdot \mathbf{k})}$ , which synchronizes the learning dynamics of parameters in $\lambda$ with model parameters in $Q, K$ projection. Moreover, to assign $\lambda$ an initialization value $\lambda_{\text{init}}$ and enables it to learn values smaller or larger than $\lambda_{\text{init}}$, we give the final formulation in Equation (2):
>
> $$
> \lambda = \exp(\mathbf{\lambda_{q_1}} \cdot \mathbf{\lambda_{k_1}}) - \exp(\mathbf{\lambda_{q_2}} \cdot \mathbf{\lambda_{k_2}}) + \lambda_{\text{init}},~ \mathbf{\lambda_{q_1}}, \mathbf{\lambda_{k_1}}, \mathbf{\lambda_{q_2}}, \mathbf{\lambda_{k_2}} \in \mathbb{R}^d
> $$
>
>
>
> > 2\. Why in Equation (3) there is a term $1 - \lambda_{\text{init}}$ for each head?
>
> The related part in Equation (3) is:
>
> $$
> \\overline{{\\mathrm{head} _ i}} = (1 - \\lambda _ {\\text{init}}) \\cdot \\operatorname{LN}( \\mathrm{head} _ i ),~ i \in [1, h]
> $$
>
> The multiplier $(1 - \lambda_{\text{init}})$ can be considered as the initialization of the scale parameter of $\operatorname{LN}$, designed to align the gradients with Transformer. At the early training stage,  with differential attention defined in Equation (1), for a token $x$ in $\mathrm{head}_i$ we have:
>
> $$
> \\cfrac{||x|| _ 2}{\\sqrt{d}}=\\Theta(1-\\lambda _ {\\text{init}})
> $$
>
> Consider the gradient of $\operatorname{LN}$:
>
> $$
> (1 - \\lambda _ {\\text{init}}) \\cdot \\cfrac{\\partial \\text{LN}(x)}{\\partial x} = \\Theta((1 - \\lambda _ {\\text{init}}) \\cdot \\cfrac{1}{||x|| _ 2 \\ / \\sqrt{d}}) = \\Theta(\\cfrac{1 - \\lambda _ {\\text{init}}}{1-\\lambda _ {\\text{init}}}) = \\Theta(1)
> $$
>
> With the multiplier $(1 - \lambda_{\text{init}})$, we counteract the gradient factor $\frac{1}{1-\lambda_{\text{init}}} > 1$, better aligning gradients of Diff Transformer with Transformer to avoid hyperparameter adjustments.
>
> The detailed gradient derivation can be found in Appendix E.
>
>
>
> > 3\. Why the RMSNorm for each head is so important for DIFF Transformer stability?
>
> The importance of headwise RMSNorm can be explained from two aspects:
>
> - The sum of attention scores is $(1 - \lambda)$ in DIFF Transformer, while it is $1$ in Transformer. The RMSNorm normalizes the output value of differential attention and ensures each token in each head has moderate magnitude.
>
> - We find that the headwise RMSNorm plays an important role in sparse attention. As differential attention exhibits sparse attention pattern, the variance in output across different tokens and different heads can be larger than in the conventional attention. Under this circumstance, we find normalizing different tokens in different heads to the same scale is beneficial.

---

### Public Comment · ~Minh_Nguyen_Hoang1 · 2024-11-25
**Comparison with cosine-based transformer**

Dear authors, have you considered comparing the performance of your work to cosine-based transformer. While the theoretical motivation coming from the differential amplifiers used in signal processing, the differential attention has the output range of $[-\lambda, 1]$, in contrast to the normal softmax-based attention only support positive similarity. I would love to see some comparision results between your proposed method and previous cosine-based attention methods or any discussion on how this method can surpassed cosine-based methods.

---

> ### Author Response · Authors · 2024-11-26
>
> Hi Minh Nguyen Hoang,
>
> In reviewing previous cosine-based attention approaches, Cosformer [1] introduces a cos-based re-weighting mechanism as a relative positional encoding to enforce locality in linear attention. SwinV2 [2] applies the cosine operation to attention logits before softmax to stabilize training.
>
> The goal of the cosine operation is to promote locality or stabilize training. Directly replacing softmax with a cosine operation in attention usually leads to worse performance compared to conventional attention.
>
> [1] cosformer: Rethinking softmax in attention. Qin, et al. 2022.
>
> [2] Swin transformer v2: Scaling up capacity and resolution. Liu, et al. 2022.

---

### Public Comment · ~Xinyu_Zhao3 · 2024-11-28
**Implementation details of Signal-to-Noise Ratio**

Dear authors, for calculating the Signal-to-Noise Ratio in Fig1 and Attention Noise in Table 3, could you please explain the attention weights come from which tokens? E.g. the query tokens/the first generated token/the first answer-relevant token/all answer tokens. Will there be any difference in Attention Noise for different tokens?

---

> ### Author Response · Authors · 2024-11-28
>
> Hi Xinyu Zhao,
>
> In the paper, the attention weights come from all answer tokens. In the key information retrieval task, we observe that all answer tokens are answer-relevant tokens. We find averaging the weights across all answer tokens results in more stable and accurate outcomes. Calculating with the first answer token can yield similar conclusions as in the paper.

---

### Public Comment · ~Kebin_Wu2 · 2024-12-04
**Other option to cancel noise**

Dear authors,

Thank you for this interesting paper.  I have two questions and would appreciate if you can provide your insights on these.
1.   since the attentions on the irrelevant tokens are considered as noise,  why don't we use addition instead of subtraction to do the noise cancellation?   the motivation is that we can improve the signal to noise ratio with the simple averaging in the field of signal processing, where random noise after adding is still low, but the signal itself can be enhanced?

2.  The differential transformer proposed here is validated with decoding-only LLM in this paper if I am not mistaken,  have you tried applying this to encoder-based models, such as the text encoder used in the CLIP model? Thank you.

---

### Meta-Review · Area_Chair_JAaJ · 2024-12-19

**Metareview:**

This paper proposes a simple (in principle) modification to the attention mechanism, coined differential attention that allows to improve the signal to noise ration in the attention maps for model with very long context windows. The proposed approach is simple, plug-in to all modern transformer-based models and shows good performance on all tasks. Given the positive reviews and interesting and simple research findings, this paper is a clear accept. Given the broad impact (generic architecture improvement) this could be an oral presentation at ICLR 2025.

**Additional Comments On Reviewer Discussion:**

The reviewers unanimously rated the paper positively.

---

### Decision · Program_Chairs · 2025-01-22

Accept (Oral)